# Recovery Time of Electrical Sensory, Motor, and Pain Thresholds: A Pilot Study Towards Standardization of Quantitative Sensory Testing in Healthy Population

**DOI:** 10.3390/healthcare13192492

**Published:** 2025-10-01

**Authors:** Izarbe Ríos-Asín, Miguel Malo-Urriés, Jorge Pérez-Rey, Marta García-Díez, Lucía Burgos-Garlito, Elena Bueno-Gracia

**Affiliations:** PhysiUZerapy Health Sciences Research Group, Health Sciences Faculty, Department of Physiatry and Nursing, University of Zaragoza, 50009 Zaragoza, Spain; irios@unizar.es (I.R.-A.); jorge.perez@unizar.es (J.P.-R.); ebueno@unizar.es (E.B.-G.)

**Keywords:** electrical threshold testing, QST, recovery time, electrical stimulation, somatosensory function, sensory testing

## Abstract

Background/Objectives: Electrical threshold testing (ETT) offers a promising method for assessing somatosensory function. Despite its growing use, fundamental aspects such as the physiological recovery time required between repeated threshold measurements remain poorly understood. This gap is critical when evaluating sensory, motor, or pain thresholds (EST, EMT, EPT) in pre–post designs or rapid intra-session protocols. The aim is to investigate the short-term recovery dynamics of electrical thresholds following electrical threshold testing, and to determine the minimum interval required for values to return to a stable baseline. Methods: In this pilot, repeated-measures study, 10 healthy adults (20 upper limbs) underwent three progressive stimulation trials (sensory, motor, and pain). Electrical thresholds were assessed at fixed recovery intervals (0–120 s), with duplicate measurements at each time point. Stability was defined as the absence of significant differences between repeated measures. Results: EST stabilized rapidly after sensory or motor stimulation, showing no significant differences beyond 0 and 15 s, respectively. Within pain stimulation, EST recovered at 60 s. EMT showed immediate recovery with motor stimulation and required longer recovery with pain stimulation, with stabilization observed at 90 s. EPT exhibited the highest variability, with the smallest time-dependent differences observed immediately after the first assessment. Conclusion: Recovery time after electrical stimulation varies by threshold type and intensity of the stimuli. EST and EMT can be reliably reassessed immediately after sensory and motor stimulation, respectively. However, when stimulation reaches EPT level, EST requires 60 s to recover and EMT needs 90 s. EPT demonstrates higher variability, indicating the need for further investigation. These findings support the implementation of standardized recovery intervals in ETT and underscore the importance of interpreting EPT results with caution during rapid assessments.

## 1. Introduction

Quantitative Sensory Testing (QST) is a valuable tool commonly used to evaluate somatosensory function and pain processing [1]. It includes various modalities—such as thermal, mechanical, and pressure stimuli—that allow the quantification of sensory thresholds and responses. In this context, electrical threshold testing (ETT) has emerged as a QST method for assessing sensory, motor and pain-related mechanisms. ETT involves the measurement of different electrical thresholds by applying controlled electrical stimulation. These thresholds include the electrical sensory threshold (EST)—the minimum current intensity needed to induce conscious perception of electrical stimulation; the electrical motor threshold (EMT)—the current intensity required to elicit a visible motor response in the muscles; and the electrical pain threshold (EPT)—the minimum intensity of current that induces the first perceptible pain sensation in the participant. Compared to traditional thermal and mechanical tests, ETT has shown greater sensitivity in detecting sensory impairments and may represent a more accessible approach due to the availability of equipment [2,3,4]. Furthermore, it offers higher objectivity, greater reproducibility, and easier standardization across sessions and examiners [5]. Unlike mechanical or thermal stimuli, which are more susceptible to contextual factors and examiner-dependent variability, electrical stimulation enables the precise quantification of sensory and motor responses across a wide range of stimulation intensities [6].

Despite these advantages, some fundamental methodological aspects of electrical QST remain insufficiently understood [7]. A particularly relevant but understudied parameter is the recovery time required between consecutive threshold assessments, that is, the interval needed between two stimulations to obtain two identical measurements. This factor is critical because repeated electrical stimulation with low-frequency biphasic currents can transiently alter the excitability of neural pathways, resulting in increased electrical thresholds in subsequent measurements. Several mechanisms may account for this effect. Neural accommodation, for instance, occurs when repeated stimulation dismisses the response, leading to peripheral fatigue and requiring higher stimulus intensities to elicit equivalent sensory or motor responses [8]. In addition, the short-term analgesic effects of low-frequency biphasic stimulation are primarily explained by the gate control theory, whereby stimulation of large diameter Aβ fibers inhibits the retransmission of the painful stimuli carried by Aδ and C fibers [9]. As this inhibitory modulation persists temporarily, higher stimulus intensities may be necessary to reproduce the same response immediately after a previous stimulation. Furthermore, central processes such as temporal summation, where repeated or continuous noxious stimuli results in increased perceived pain despite the same intensity of the stimuli, may also influence ETT [10]. Other central mechanisms might be considered such as endogenous opioid-mediated analgesia, serotonergic modulation, and reductions in excitatory neurotransmitters such as aspartate and glutamate within the medulla [11].

Without a clear understanding of how quickly thresholds return to baseline after stimulation, the validity of repeated-measurements designs—commonly used in standardized QST protocols [12]—may be compromised. This limitation is particularly relevant in ETT protocols, which have been less extensively studied and standardized compared to traditional QST. Addressing this gap may also help to ensure consistency in pre–post assessments, intra-session monitoring, and longitudinal studies, thereby advancing the application of ETT while strengthening its methodological rigor.

To this end, the present pilot study aimed to investigate the short-term recovery dynamics of electrical thresholds following neuromuscular electrical stimulation. Using a repeated-measures design with systematically varied recovery intervals (0 to 120 s), we sought to determine the minimum time required for thresholds to stabilize and whether this recovery pattern differs across threshold types. Our findings are intended to inform the development of standardized protocols for reliable intra-session ETT assessment and support its broader application in experimental and clinical settings.

## 2. Materials and Methods

### 2.1. Study Design

A pilot observational, descriptive, cross-sectional, and prospective study was conducted. The study was approved by the Research Ethics Committee of the Autonomous Community of Aragón (CEICA) (C.I. PI24/248) on 24 May 2024 and received authorization for the processing of personal data from the Data Protection Unit of the University of Zaragoza (CUSTOS), with reference number RAT 2024-142. All participants provided written informed consent in accordance with the Declaration of Helsinki [13].

### 2.2. Participants

A convenience sample of 10 healthy adult volunteers (5 males, 5 females) was recruited. Each participant contributed both upper limbs to the study, yielding a total of 20 extremities. To be eligible, participants were required to be aged 18 years or older, to have the ability to communicate effectively, and to understand the procedures performed. Exclusion criteria included a history of chronic disorders—such as endocrine, neurological, psychiatric, urogenital, musculoskeletal, dermatological diseases, or chronic pain syndromes—or any other factor that could interfere with the results.

Given the exploratory nature of the study and the absence of prior data on the intra-time point stability of electrical thresholds following neuromuscular stimulation, this sample was considered appropriate for a pilot design [14,15]. The chosen size aligns with established recommendations for pilot studies, allowing for effect size estimation and feasibility assessment while minimizing resource demands and participant burden [16,17]. The results obtained from this preliminary dataset will serve to guide formal power analyses and inform future confirmatory research with larger sample sizes.

### 2.3. Electrical Threshold Testing

Each participant underwent a single assessment session, during which they completed a brief demographic questionnaire. The primary variables measured were the following: the EST, defined as the minimum current intensity in mA required to induce conscious sensory perception; the EMT, defined as the minimum intensity in mA needed to elicit a visible muscle contraction; and the EPT, defined as the lowest current intensity in mA that produced a clearly perceptible painful sensation [18].

All thresholds were measured using low-frequency symmetrical biphasic current with GYMNA MYO 200 electrotherapy device (No. 320.290). Disposable, squared, 25 cm^2^ electrodes were used and the stimulation parameters were set at a frequency of 100 Hz, a pulse width of 100 μs, and an increment rate of 1 mA per s. Participants were positioned supine with the dominant forearm in supination [19]. The forearm was selected as a representative site for sensory evaluation. Electrodes were placed on the anterior aspect of the forearm, aligned longitudinally along the wrist flexor muscle group. The distal electrode was positioned 4 cm proximal to the wrist joint line, while the proximal electrode was placed 4 cm distal to the elbow crease [20]. To minimize inter-examiner variability, all electrode placement and removal procedures were performed by the same trained investigator.

### 2.4. Assessment Procedure

Figure 1 represents the procedure followed. Three progressive stimulation trials were conducted to assess threshold stability over time, with an inter-test time of 10 min.

Sensory Threshold Test (STT): Electrical stimulation was increased until the EST was reached. Pairs of two consecutive measurements were recorded at each of the following recovery intervals: 0 s, 15 s, 30 s, and 60 s.

Motor Threshold Test (MTT): Electrical stimulation was increased until the motor visible response. Pairs of EST and EMT values were recorded at 0 s, 15 s, 30 s, and 60 s.

Pain Threshold Test (PTT): Electrical stimulation was increased until the participant reported the first-pain sensation. Two consecutive measurements were recorded at each of the following recovery intervals: 0 s, 15 s, 30 s, 60 s, 90 s, and 120 s.

At each time point, pairs of two measurements were recorded. Recovery intervals of 3 min were allowed between consecutive measurements to avoid carryover effects. All assessments were conducted in a controlled environment by the same experienced examiner to ensure procedural consistency.

### 2.5. Statistical Analysis

A descriptive and inferential statistical analysis was conducted using IBM SPSS Statistics (version 29.0, IBM Corp., Armonk, NY, USA). The main objective was to determine the minimum time required for the recovery of electrical thresholds following stimulation, based on the stabilization of repeated measures at each time point. Recovery was defined operationally as the absence of statistically significant differences (*p* ≥ 0.05) between two consecutive threshold measurements obtained at the same time point.

For each test a series of paired comparisons were performed between the first and second measurements obtained at each recovery time (0 s, 15 s, 30 s, 60 s, 90 s, and 120 s, depending on the test). The Shapiro–Wilk test was applied to assess the normality of the distribution of threshold values. When the data met parametric assumptions, paired-samples *t*-tests were used; otherwise, the non-parametric Wilcoxon signed-rank test was applied. A *p*-value < 0.05 was considered indicative of statistically significant differences. No averaging was conducted between repeated measures, as the study’s aim was specifically to evaluate intra-time point stability as an indicator of threshold recovery.

Additionally, graphical analysis was performed to visualize the differences between the two measures at each time point. The evolution of threshold values over time was plotted for each participant, and difference scores (second measurement minus first measurement) were computed and represented to identify time points of stabilization (i.e., when the difference approached zero with low dispersion).

## 3. Results

### 3.1. Descriptive Analysis of the Sample

The sample consisted of 10 participants. Table 1 presents the analysis of the descriptive variables. The mean age of the sample was 23.80 ± 3.66 years, with an average height of 168.50 ± 12.60 cm and an average weight of 66.00 ± 17.57 kg. Of the participants, 50% were female and 100% were right-handed.

### 3.2. Sensory Threshold Testing

Table 2 presents the results of the STT measurements taken at 0, 15, 30, and 60 s after sensitive stimulation. No statistically significant differences were found between the first and second measurements at any time point (*p* > 0.05 in all comparisons), indicating a rapid stabilization of EST. These findings suggest that the EST recovers quickly and remains stable within the first minute after application. Figure 2a represents in a graphic the difference between the two consecutive measurements at each time point.

### 3.3. Motor Threshold Testing

Table 3 shows the results for EST and EMT measurements during MTT. For EST, a statistically significant difference was found at 0 s (*p* = 0.014), indicating that sensory thresholds had not yet stabilized immediately after stimulation. However, no significant differences were observed at 15, 30, or 60 s (*p* > 0.05), and the mean differences were minimal (≤0.10). These results suggest that EST recover and stabilize rapidly within the first 15 s following the motor-level stimulation.

Regarding EMT, no statistically significant differences were observed at any time point (all *p* > 0.05), suggesting that motor thresholds remained generally stable throughout the recovery intervals. Although differences at 0 and 15 s approached the conventional threshold for significance (*p* = 0.053 and *p* = 0.079, respectively), these values did not reach statistical significance, indicating that the variability observed at these early intervals may not be clinically relevant. From 30 s onwards, differences between repeated measures were minimal (e.g., 0.03 ± 0.53 at 30 s), reinforcing the overall stability of EMT values over time. Figure 2b represents in a graphic the difference between the two consecutive measurements at each time point for motor stimulation.

### 3.4. Pain Threshold Testing

As shown in Table 4, statistically significant differences were found between the two EST at 0 s (*p* < 0.001), 15 s (*p* = 0.034), and 30 s (*p* = 0.002), suggesting incomplete recovery of the sensory threshold during the early time points of PTT. At 60 s, however, the difference was reduced to 0.30 ± 0.82 and no longer statistically significant (*p* = 0.117), indicating progressive stabilization. These results suggest that after painful-level stimulation, EST requires more time to recover than in the previous tests.

Significant differences were observed at the range of 0–60 s for EMT (all *p* ≤ 0.022). Although the magnitude of difference decreased progressively—from 1.93 ± 1.10 at 0 s to 0.40 ± 0.72 at 60 s—these results suggest that EMT remain unstable for longer durations following painful-level stimulation. For 90 s and 120 s, no significant differences were obtained in EMT (*p* > 0.408), suggesting that full recovery may require more than 1 min.

For EPT, no significant difference was obtained at 0 s (*p* = 0.428), 30 s (*p* = 0.238) or 90 s (*p* = 0.324), but statistically significant changes were observed at 15 s (*p* = 0.017), 60 s (*p* = 0.034) and 120 s (*p* < 0.001). The irregular pattern suggests a variable response in pain perception during repeated measurements. These findings imply that although the EPT appears relatively stable at baseline, transient fluctuations may occur during the recovery window, particularly between 15 and 60 s. Figure 2c shows the differences between measurements A-B for EST, EMT and EPT in the range of 0–120 s.

## 4. Discussion

This pilot study aimed to determine the minimum recovery time required for the electrical thresholds following different ETT. Recovery was defined as the absence of statistically significant differences between two pairs of measurements obtained with different time intervals. The results provide preliminary evidence that the recovery time of electrical thresholds differ markedly depending on the type of stimulation applied. Specifically, EST values stabilized almost immediately after STT, while EMT required up to 60 s to return to stable values following MTT. After PTT, both EST and EMT showed delayed recovery, with stabilization occurring beyond 60–90 s. EPT exhibited a more irregular and inconsistent pattern, with statistically significant fluctuations at various time points, suggesting that pain perception may be more susceptible to cognitive, attentional, or neurophysiological variability.

The temporal differences observed in threshold recovery across stimulation levels appear to reflect distinct underlying neurophysiological mechanisms. Results showed that EST stabilized rapidly after STT and MTT, with no statistically significant differences beyond 15 s. In contrast, EST recovery following PTT was delayed, with significant variability persisting up to 30 s and only reaching stability by around 60 s. These findings support the hypothesis that the intensity and type of preceding stimulation influence the excitability and recovery kinetics of the sensory system. In the case of low-intensity sensory input, afferent pathways, primarily mediated by Aβ fibers, are activated in a non-fatiguing and non-adaptive manner, allowing for quick return to baseline. However, painful stimuli likely induce central and peripheral modulation mechanisms, including short-term sensitization of nociceptive pathways and increased excitability of dorsal horn neurons, which may temporarily alter subsequent sensory processing and delay recovery of thresholds [21,22].

The EMT demonstrated slower recovery dynamics compared to EST. Even after MTT, differences at 0 and 15 s approached significance, suggesting early-phase variability. Following PTT, EMT remained significantly altered up to 60 s, stabilizing only beyond 90 s. This prolonged instability may be explained by transient changes in spinal excitability and motor unit recruitment thresholds, potentially related to segmental reflex modulation or the post-activation depression of alpha motoneurons. Additionally, the overlap of sensory and motor fibers (Aβ and Aα) in mixed nerves could contribute to delayed normalization of motor response thresholds when exposed to noxious stimuli [23].

The most complex behavior was observed in EPT values, which displayed non-linear and irregular patterns across time points. Statistically significant differences appeared inconsistently (at 15, 60, 120 s) despite stability at other moments (at 0, 30, 90 s). This suggests that pain perception is highly variable, likely influenced by cognitive and emotional modulation as well as intrinsic fluctuations in central pain processing [24]. The unpredictability of EPT stabilization contrasts with the more systematic patterns of EST and EMT and may reflect the dynamic interplay between peripheral nociceptive input and central modulation mechanisms, such as descending inhibition and temporal summation [22]. The progressive delay in recovery observed from sensory–motor–pain thresholds align with increasing activation of more complex and labile neurophysiological systems. While EST appears to reflect a relatively stable sensory channel under non-noxious conditions, EMT and EPT are more susceptible to neuroplastic changes, inhibitory-excitatory imbalances, and perceptual variability, particularly following intense or painful stimuli.

To date, no previous studies have examined the short-term recovery times of electrical thresholds following different levels of electrical stimulation to ensure consistency in consecutive measurements. Previous research has instead focused on broader aspects of reliability. For instance, Streuli et al. (2023) reported intra- and inter-session reliability of EST and EPT during cutaneous and intramuscular stimulation using needle electrodes in the lumbar region [15]. Their findings demonstrated high intra-session and moderate inter-session reliability, thereby supporting the use of ETT. However, their repeated-measures design does not directly compare with the present study, as it did not address the within-minute recovery of EST, EMT, and EPT. Similarly, Gaudreault et al. (2015) showed that EST is a stable and reproducible measure under standardized conditions, reporting high inter-day reliability in healthy volunteers, with Intraclass Correlation Coefficients (ICCs) ranging from 0.66 to 0.95 [25]. However, their design was centered on day-to-day reproducibility, with assessments spaced by 24 h or more, so no comparable with the present results. Additional support for ETT reliability is provided by Furuse et al. (2019), who investigated EST and EPT in the mandibular mucosa using the Neurometer EST/C^®^ system [26]. Their results confirmed high test–retest reliability, particularly for the mental foramen, with ICCs above 0.8, reinforcing the stability of electrical thresholds under controlled conditions. Xia et al. (2020) further contributed by assessing EST with circular pin electrodes, reporting moderate inter-session reliability (ICC = 0.595), and noticing physiological changes in superficial blood flow and skin temperature [18]. Such observations highlight autonomic changes that could affect short-term stability, a factor addressed directly in the present design. All these previous studies represent important advances towards ETT standardization providing evidence for its inter- and intraday reliability. Nonetheless, none addressed immediate reproducibility within the same session across different stimulation intensities. The present study extends this line of research by focusing specifically on short-term recovery, a perspective that is critical for protocols involving consecutive threshold evaluations, where residual effects from prior stimulation could otherwise confound results.

The present findings have clear implications for both clinical application and the methodological design of studies involving electrical stimulation and QST. Threshold-based protocols are widely used in physiotherapy and neuroscience to evaluate neural excitability, titrate therapeutic intensities, or monitor changes in sensory and motor function. However, the reliability of these procedures depends critically on the stability of measurements and the avoidance of residual effects from prior stimuli. These results indicate that the recovery time required to obtain stable threshold values is not uniform across all conditions but rather depends on the type of stimulation applied and the physiological system being assessed. For STT, consecutive measurements of EST may be performed. After MTT, to assess EST it is necessary to wait 15 s, while EMT can be performed immediately. For PTT, EST and EMT require 60 s and 90 s, respectively, to stabilize, while EPT remains inconsistent. This highlights the increased vulnerability of nociceptive processing to contextual and endogenous modulation and reinforces the need for cautious interpretation of EPT measurements. From a methodological standpoint, these findings underscore the importance of controlling for inter-stimulus recovery times in repeated-measures protocols, particularly those involving excitability testing, pain perception, or dose titration procedures. Failing to account for incomplete recovery may result in measurement error, false variability, or overestimation of treatment effects. The incorporation of structured recovery intervals—adapted to the stimulation intensity and threshold type—should therefore be considered a methodological requirement in both research and clinical protocols involving electrical stimulation.

While this was a pilot study with inherent exploratory value, it incorporated several methodological strengths—such as a within-subject design, tightly controlled experimental conditions (including standardized posture, electrode placement, and a single trained examiner), and repeated measurements at fixed time points to assess intra-time point stability. However, it also presents several limitations. First, the sample size was limited to 10 participants (20 extremities), which may constrain statistical power and limit the generalizability of the findings. Although our sample was evenly distributed between male and female participants, subgroup analyses by gender were not performed due to the limited sample size of this pilot study. Future studies with larger populations will be designed to explore potential gender-related differences in threshold recovery dynamics. Second, the study population consisted exclusively of young, healthy adults, reducing the applicability of the results to clinical populations or individuals with altered sensory or motor processing. Third, stimulation was applied using a single parameter configuration (100 Hz, 100 μs, rectangular biphasic waveform), so it remains uncertain whether other settings would produce similar recovery dynamics. In addition, EMT was determined by visual muscle contraction rather than by other objective methods, such as electromyography. Also, intra-rater reliability and long-term reproducibility were not assessed. Finally, the short time intervals between repeated measurements may have introduced minor habituation or attentional bias, particularly in PTT.

## 5. Conclusions

This pilot study provides preliminary insights into the recovery dynamics of fibers following neuromuscular stimulation. The findings suggest that 0 s might be enough to recover after EST stimulation, 15 s for EMT stimulation, and around 60 s for EPT stimulation. Even though the results support the feasibility of using short recovery intervals for ETT, particularly for EST and EMT, caution is suggested when interpreting EPT values due to their inconsistent stabilization. These findings may help the development of more standardized protocols for ETT, although the conclusions must be interpreted considering the study’s limitations. Further research with larger and more diverse populations, including clinical groups, is needed to confirm these observations and to define recommendations for appropriate recovery intervals in different contexts.

## Figures and Tables

**Figure 1 healthcare-13-02492-f001:**
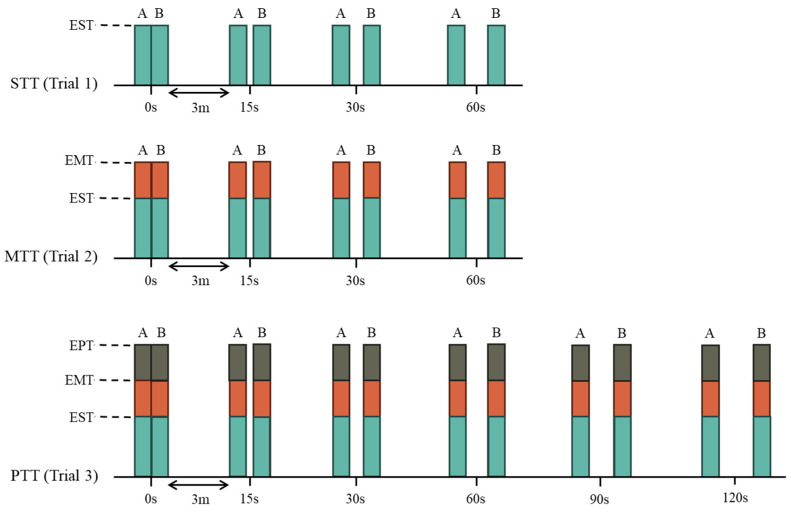
Assessment Procedure. A: first measurement of the pair; B: second measurement of the pair; EMT: Electrical Motor Threshold; EPT: Electrical Pain Threshold; EST: Electrical Sensory Threshold; MTT: Motor Threshold Test; PTT: Pain Threshold Test; STT: Sensory Threshold Test.

**Figure 2 healthcare-13-02492-f002:**
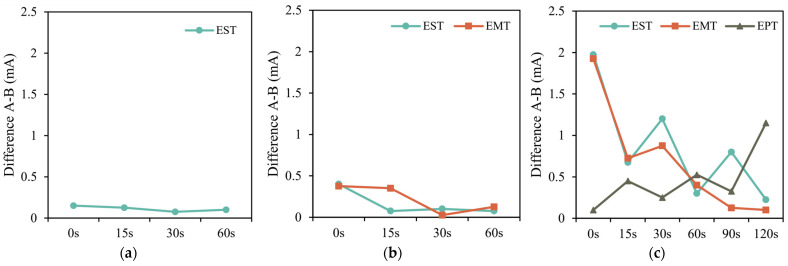
Differences between two consecutive measurements at each time point for sensitive, motor and pain stimulation. Values are expressed as difference between two consecutive measurements (A–B), presented in mA. (**a**) Sensory Threshold Testing; (**b**) Motor Threshold Testing; (**c**) Pain Threshold Testing. EMT: Electrical Motor Threshold; EPT: Electrical Pain Threshold; EST: Electrical Sensory Threshold.

**Table 1 healthcare-13-02492-t001:** Descriptive analysis of the sample.

Outcome	Mean/AF	SD/%
Age (years)	23.80	3.66
Height (cm)	168.50	12.60
Weight (kg)	66.00	17.57
Sex (women)	10	50.00
Laterality (right)	20	100.00

Categorical variables are expressed as absolute frequencies (AF) and percentages (%) within each group. Quantitative variables are expressed as mean and standard deviation (SD).

**Table 2 healthcare-13-02492-t002:** Sensory threshold testing: comparison between two consecutive measurements at each time point.

Outcome	Time	Measurement A (mA)	Measurement B (mA)	Difference (mA)	95% CI	*p*-Value
EST0	0 s	9.72 ± 3.20	9.57 ± 2.97	0.15 ± 0.56	[−0.11, 0.41]	0.272
EST15	15 s	9.60 ± 3.09	9.72 ± 3.09	0.13 ± 0.43	[−0.07, 0.32]	0.197
EST30	30 s	9.38 ± 2.65	9.30 ± 2.84	0.08 ± 0.57	[−0.34, 0.19]	0.432
EST60	60 s	9.20 ± 2.78	9.10 ± 2.69	0.10 ± 0.62	[−0.39, 0.19]	0.479

Values are expressed as mean ± standard deviation. The statistical comparison was performed between the first and second measurements at each time point. A *p*-value < 0.05 was considered statistically significant. 95% CI: 95% Confidence Interval; EST0: Electrical Sensory Threshold with an interval recovery of 0 s.

**Table 3 healthcare-13-02492-t003:** Motor threshold testing: comparison between two consecutive measurements at each time point.

Outcome	Time	Measurement A (mA)	Measurement B (mA)	Difference (mA)	95% CI	*p*-Value
EST0	0 s	9.15 ± 2.77	9.55 ± 3.00	0.40 ± 0.66	[0.09, 0.71]	0.014 *
EST15	15 s	9.65 ± 2.99	9.72 ± 3.16	0.08 ± 1.10	[−0.44, 0.59]	0.630
EST30	30 s	9.80 ± 3.09	9.90 ± 3.09	0.10 ± 1.07	[−0.40, 0.60]	0.972
EST60	60 s	9.85 ± 2.97	9.93 ± 3.00	0.08 ± 0.54	[−0.18, 0.33]	0.545
EMT0	0 s	16.12 ± 3.29	16.50 ± 2.99	0.38 ± 0.96	[−0.07, 0.82]	0.053
EMT15	15 s	16.20 ± 3.14	16.55 ± 3.26	0.35 ± 0.83	[−0.04, 0.74]	0.079
EMT30	30 s	16.43 ± 3.11	16.45 ± 3.19	0.03 ± 0.53	[−0.22, 0.27]	0.805
EMT60	60 s	16.50 ± 3.18	16.38 ± 3.34	−0.13 ± 0.58	[−0.40, 0.15]	0.388

Values are expressed as mean ± standard deviation. The statistical comparison was performed between the first and second measurements at each time point. A *p*-value < 0.05 was considered statistically significant. 95% CI: 95% Confidence Interval; EMT0: Electrical Motor Threshold with an interval recovery of 0 s; EST0: Electrical Sensory Threshold with an interval recovery of 0 s. * *p* < 0.05.

**Table 4 healthcare-13-02492-t004:** Pain threshold testing: comparison between two consecutive measurements at each time point.

Outcome	Time	Measurement A (mA)	Measurement B (mA)	Difference (mA)	95% CI	*p*-Value
EST0	0 s	9.88 ± 3.13	11.85 ± 3.60	1.98 ± 1.33	[1.35, 2.60]	<0.001 **
EST15	15 s	10.50 ± 3.15	11.18 ± 3.18	0.68 ± 1.32	[0.06, 1.29]	0.034 *
EST30	30 s	10.03 ± 3.16	11.22 ± 3.21	1.20 ± 1.48	[0.51, 1.89]	0.002 **
EST60	60 s	10.95 ± 3.34	11.25 ± 3.53	0.30 ± 0.82	[−0.08, 0.68]	0.117
EST90	90 s	10.30 ± 2.92	11.10 ± 3.64	0.80 ± 1.76	[−0.02, 1.62]	0.056
EST120	120 s	10.75 ± 3.27	10.98 ± 3.29	0.23 ± 0.91	[−0.20, 0.65]	0.283
EMT0	0 s	16.50 ± 3.20	18.43 ± 3.43	1.93 ± 1.10	[1.41, 2.44]	<0.001 **
EMT15	15 s	17.07 ± 3.18	17.80 ± 3.02	0.73 ± 0.77	[0.37, 1.08]	<0.001 **
EMT30	30 s	17.15 ± 3.29	18.02 ± 3.15	0.88 ± 0.79	[0.50, 1.25]	<0.001 **
EMT60	60 s	17.90 ± 3.58	18.30 ± 3.54	0.40 ± 0.72	[0.06, 0.74]	0.022 *
EMT90	90 s	17.23 ± 3.11	17.35 ± 3.34	0.13 ± 0.74	[−0.22, 0.47]	0.460
EMT120	120 s	17.95 ± 3.25	18.05 ± 3.36	0.10 ± 0.53	[−0.15, 0.35]	0.408
EPT0	0 s	23.55 ± 3.72	23.65 ± 3.52	0.10 ± 0.55	[−0.16, 0.36]	0.428
EPT15	15 s	23.57 ± 3.43	24.02 ± 3.33	0.45 ± 0.74	[0.10, 0.80]	0.017 *
EPT30	30 s	24.35 ± 3.40	24.60 ± 3.33	0.25 ± 0.90	[−0.17, 0.67]	0.238
EPT60	60 s	25.07 ± 3.73	25.60 ± 3.69	0.53 ± 1.01	[0.05, 1.00]	0.034 *
EPT90	90 s	25.63 ± 3.70	25.95 ± 3.89	0.33 ± 1.44	[−0.35, 1.00]	0.324
EPT120	120 s	26.45 ± 4.12	27.60 ± 4.57	1.15 ± 0.25	[0.63, 1.67]	<0.001 **

Values are expressed as mean ± standard deviation. The statistical comparison was performed between the first and second measurements at each time point. A *p*-value < 0.05 was considered statistically significant. 95% CI: 95% Confidence Interval; EMT0: Electrical Motor Threshold with an interval recovery of 0 s; EPT0: Electrical Pain Threshold with an interval recovery of 0 s; EST0: Electrical Sensory Threshold with an interval recovery of 0 s. * *p* < 0.05; ** *p* < 0.01.

## Data Availability

The original contributions presented in this study are included in the article. Further inquiries can be directed to the corresponding author(s).

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
