# Peer review of "Recovery Time of Electrical Sensory, Motor, and Pain Thresholds: A Pilot Study Towards Standardization of Quantitative Sensory Testing in Healthy Population"

_healthcare, 2025, doi:10.3390/healthcare13192492_

Round 1
Reviewer 1 Report
Comments and Suggestions for Authors
The title of the article I received for the review "Recovery Time of Electrical Sensory, Motor, and Pain Thresholds: A Pilot Study to Support the Standardization of QST Protocols" reflects the experimental work that was carried out. As a pilot study, it has a small sample, which is also stated in the text of the discussion by the researchers themselves, and the results obtained should certainly be further examined in future work on a larger sample. The article is very systematically and clearly written. The article contributes to a more adequate understanding of the possibilities and limitations of the application of QST in future clinical research and practical application.
In the description of the group of respondents, although it goes without saying, due to the importance of these data, it would be good also to mention dermatological diseases and chronic pain syndromes as exclusion criteria.
Methodology and statistical methods are clearly described.
The results that are shown are clearly presented and clear explanations are given
Although half of the respondents in the research were female and half were male, the results and discussion did not mention the results obtained in relation to gender, nor were they further commented, either in terms of the obtained or expected results and their explanation. If these results will be shown in another article, it should be stated It can be written as future work.
The conclusions are clear and stem from the obtained results
For reference 3, when citing the journal, write a capital letter for the journal Pain.
I consider the paper suitable for publication with minor changes.
Author Response
1. Summary |
|
|
“Pain Thresholds: A Pilot Study to Support the Standardization of QST Protocols" reflects the experimental work that was carried out. As a pilot study, it has a small sample, which is also stated in the text of the discussion by the researchers themselves, and the results obtained should certainly be further examined in future work on a larger sample. The article is very systematically and clearly written. The article contributes to a more adequate understanding of the possibilities and limitations of the application of QST in future clinical research and practical application.
Author’s Response
We sincerely thank the reviewer for their careful reading of our manuscript and their constructive comments. We appreciate the positive feedback and have addressed all the suggested points to improve the quality and clarity of the article. Below, we provide a detailed response to each comment, including the changes made in the revised version.
|
||
2. Point-by-point response to Comments and Suggestions for Authors
|
||
Comment: In the description of the group of respondents, although it goes without saying, due to the importance of these data, it would be good also to mention dermatological diseases and chronic pain syndromes as exclusion criteria. |
||
Response: Thank you for this suggestion. We agree that explicitly including these conditions improves the clarity of the inclusion and exclusion criteria. We have updated the Participants section (Section 2.2, lines 99-102) to include dermatological diseases and chronic pain syndromes as exclusion criteria. Revised text: "Exclusion criteria included a history of chronic disorders—such as endocrine, neurological, psychiatric, urogenital, musculoskeletal, dermatological diseases, or chronic pain syndromes—or any other factor that could interfere with the results." |
||
Comment: Methodology and statistical methods are clearly described. |
||
Response: We appreciate the reviewer’s positive feedback regarding the clarity of the methodology and statistical approach. |
||
Comment: The results that are shown are clearly presented and clear explanations are given |
||
Response: Thank you for your comment. We are glad that the presentation of the results was clear and well-explained. |
||
Comment: Although half of the respondents in the research were female and half were male, the results and discussion did not mention the results obtained in relation to gender, nor were they further commented, either in terms of the obtained or expected results and their explanation. If these results will be shown in another article, it should be stated It can be written as future work. |
||
Response: We appreciate the reviewer’s observation. Given the small sample size of this pilot study (n = 10), subgroup analyses by gender would lack statistical power and could be misleading. Therefore, we have added a sentence in the Discussion section (Section 4, lines 349-352) to explicitly state this limitation and to indicate that gender differences will be explored in future studies with larger samples. Change made in the manuscript: "Although our sample was evenly distributed between male and female participants, subgroup analyses by gender were not performed due to the limited sample size of this pilot study. Future studies with larger populations will be designed to explore potential gender-related differences in threshold recovery dynamics." |
||
Comment: The conclusions are clear and stem from the obtained results. |
||
Response: Thank you for recognizing that the conclusions are well-supported by our findings. |
||
Comment: For reference 3, when citing the journal, write a capital letter for the journal Pain. |
||
Response: Thank you for pointing out this typographical error. We have corrected the journal title in reference 3 so that Pain now appears with the correct capitalization. |
||
Comment: I consider the paper suitable for publication with minor changes. |
||
Response: We greatly appreciate the reviewer’s overall positive evaluation and constructive suggestions to improve the manuscript. |
Reviewer 2 Report
Comments and Suggestions for Authors
Thank you for sharing your research manuscript; it is both thorough and insightful. It addresses a crucial and essential question regarding the optimal waiting time between successive electrical threshold measurements within a session. It does so through a well-structured pilot study involving 10 healthy participants, encompassing both upper limbs (20 extremities) and repeated measures across sensory, motor, and pain trials.
The Methods section is clear and informative, outlining posture, electrode placement, and the use of “low-frequency symmetrical biphasic current.” Additionally, the definition of “recovery” provided is constructive, as it indicates no statistically significant difference between consecutive measures at specific time points. These thoughtful choices definitely enhance the clarity of your findings and contribute to the development of effective protocols.
To further strengthen your manuscript and ensure precision and reproducibility, I would like to suggest some minor edits:
1. Align the Conclusions with Table 4 regarding EPT at 120 s. In the Conclusions, you note that “EPT exhibited greater variability, although differences were less than 1 mA.” However, Table 4 states EPT120 = 1.15 ± 0.25 mA (p < 0.001). I recommend revising the sentence to reference the 1.15 mA value directly for clarity.
2. Use a single ethics approval identifier consistently. I noticed two different CEICA codes: C.I. PI24/248 in the Study Design and C.I.PI23/646 in the IRB statement. It would be beneficial to reconcile this by using one code throughout the manuscript for consistency.
3. Move the specifics of the stimulation waveform into the Methods section. The Methods currently mention “low-frequency symmetrical biphasic current,” while the Limitations section provides exact parameters (100 Hz, 100 μs). Including these specific waveform details in the Methods section would enhance the reproducibility of your setup. You could choose to keep or adjust the duplicated note in the Limitations for further clarity.
I compliment you on the consistency you've maintained with your tables and figures. The Results section effectively outlines the critical timeframes. Keeping these specific timeframes highlighted helps readers apply your guidance accurately.
For clarity and consistency, it might be beneficial to define “recovery” in terms of p-values, stating that there is no statistically significant difference between consecutive measurements at a given time point. I suggest that the authors enhance the presentation of results by adding 95% confidence intervals for the mean A–B differences at each time point to complement p-values and quantify precision, as well as considering a brief Bland–Altman agreement plot. This addition could effectively illustrate precision without altering the analysis and would offer a visual representation of agreement where “recovery” is claimed.In summary, with these revisions, the manuscript will be more precise and ready for publication, all while preserving the integrity of your design and findings. Thank you very much again for your important work, and I look forward to seeing the revised version.
Author Response
1. Summary |
|
|
Thank you for sharing your research manuscript; it is both thorough and insightful. It addresses a crucial and essential question regarding the optimal waiting time between successive electrical threshold measurements within a session. It does so through a well-structured pilot study involving 10 healthy participants, encompassing both upper limbs (20 extremities) and repeated measures across sensory, motor, and pain trials. The Methods section is clear and informative, outlining posture, electrode placement, and the use of “low-frequency symmetrical biphasic current.” Additionally, the definition of “recovery” provided is constructive, as it indicates no statistically significant difference between consecutive measures at specific time points. These thoughtful choices definitely enhance the clarity of your findings and contribute to the development of effective protocols. To further strengthen your manuscript and ensure precision and reproducibility, I would like to suggest some minor edits:
Author’s Response We would like to sincerely thank the reviewer for their careful reading of our manuscript and their constructive suggestions. We appreciate the positive evaluation and have implemented several changes to further improve the clarity, reproducibility, and precision of the study. Below, we address each comment point by point and describe the modifications made to the revised manuscript.
|
||
2. Point-by-point response to Comments and Suggestions for Authors
|
||
Comment: 1. Align the Conclusions with Table 4 regarding EPT at 120 s. In the Conclusions, you note that “EPT exhibited greater variability, although differences were less than 1 mA.” However, Table 4 states EPT120 = 1.15 ± 0.25 mA (p < 0.001). I recommend revising the sentence to reference the 1.15 mA value directly for clarity. |
||
Response: Thank you for this observation. We have revised the sentence in the Conclusions section (Section 5, lines 366-368) to accurately reflect the data presented in Table 4 and to directly include the exact value. Revised text: " Even though the results support the feasibility of using short recovery intervals for ETT, particularly for EST and EMT, caution is suggested when interpreting EPT values due to their inconsistent stabilization. " |
||
Comment: 2. Use a single ethics approval identifier consistently. I noticed two different CEICA codes: C.I. PI24/248 in the Study Design and C.I.PI23/646 in the IRB statement. It would be beneficial to reconcile this by using one code throughout the manuscript for consistency. |
||
Response: We appreciate this clarification. The correct CEICA identifier for this study is C.I. PI24/248. We have revised the Study Design section (Section 2.1, line 91) and the IRB Statement (Section 5, line 380) to ensure that only this correct code appears consistently throughout the manuscript. |
||
Comment: 3. Move the specifics of the stimulation waveform into the Methods section. The Methods currently mention “low-frequency symmetrical biphasic current,” while the Limitations section provides exact parameters (100 Hz, 100 μs). Including these specific waveform details in the Methods section would enhance the reproducibility of your setup. You could choose to keep or adjust the duplicated note in the Limitations for further clarity. |
||
Response: Thank you for this suggestion. We have moved the exact stimulation parameters into the Methods section (Section 2.3, lines 120-122), while keeping a brief reference in the Limitations section for completeness. Revised text added to Methods: "…the stimulation parameters were set at a frequency of 100 Hz, a pulse width of 100 μs, and an increment rate of 1 mA per s." |
||
Comment: I compliment you on the consistency you've maintained with your tables and figures. The Results section effectively outlines the critical timeframes. Keeping these specific timeframes highlighted helps readers apply your guidance accurately. For clarity and consistency, it might be beneficial to define “recovery” in terms of p-values, stating that there is no statistically significant difference between consecutive measurements at a given time point. I suggest that the authors enhance the presentation of results by adding 95% confidence intervals for the mean A–B differences at each time point to complement p-values and quantify precision, as well as considering a brief Bland–Altman agreement plot. This addition could effectively illustrate precision without altering the analysis and would offer a visual representation of agreement where “recovery” is claimed. |
||
Response: We agree with this valuable recommendation. Recovery is now defined in the manuscript as: Recovery was defined operationally as the absence of statistically significant differences between two consecutive threshold measurements obtained at the same time point.” to explicitly state that recovery was operationally defined as the absence of a statistically significant difference (p ≥ 0.05) between two consecutive measurements at a given time point. |
||
Comment: "I suggest that the authors enhance the presentation of results by adding 95% confidence intervals for the mean A–B differences at each time point to complement p-values and quantify precision, as well as considering a brief Bland–Altman agreement plot." |
||
Response: We thank the reviewer for this valuable suggestion. We have added 95% confidence intervals for the mean A–B differences at each time point, as recommended, to complement p-values and better quantify precision. Regarding the Bland–Altman plot, while we appreciate the importance of this analysis, the limited sample size of this pilot study (n = 10) would not allow for a robust or reliable interpretation of agreement limits. Therefore, we have opted not to include the plot in the current manuscript but will consider it for future studies with larger samples. |
Reviewer 3 Report
Comments and Suggestions for Authors
Thank you very much for the opportunity to review this manuscript.
The title of manuscripts must be without abbreviations. It is important to indicate in the title that the study was conducted on healthy population.
In lines 39-40: “However, despite its broad application, QST presents certain limitations related to reliability, standardization, and test duration” I do not understand why there are such limitations. Conducting complex testing of tactile, temperature, pain and vibration sensitivity is at least mandatory when examining patients with distal polyneuropathy. Since each test is responsible for assessing each modality separately.
In lines 40-42-“Thermal and mechanical tests are prone to contextual influences, participant variability, and sensitization effects, which may compromise result consistency”. Indeed, a number of authors believe that patients with severe pain develop hyposensitivity of the somatosensory system to nonnoxious mechanical stimuli due to central sensitization. This functional hypoesthesia is reversible and regresses after the pain subsides. This phenomenon is called non-dermatomal somatosensory deficit. Restate this information more precisely, with explanations. I have added additional sources that may be useful for your task. This is a very important section, and it makes sense to focus on it.
- Matesanz, L.; Hausheer, A.C.; Baskozos, G.; Bennett, D.L.H.; Schmid, A.B. Somatosensory and psychological phenotypes associated with neuropathic pain in entrapment neuropathy. Pain 2021, 162, 1211–1220.
- Egloff, N.; Sabbioni, M.E.; Salathé, C.; Wiest, R.; Juengling, F.D. Nondermatomal somatosensory deficits in patients with chronic pain disorder: Clinical findings and hypometabolic pattern in FDG-PET. Pain 2009, 145, 252–258.
- Mailis-Gagnon, A.; Nicholson, K. Nondermatomal somatosensory deficits: Overview of unexplainable negative sensory phenomena in chronic pain patients. Curr. Opin. Anaesthesiol. 2010, 23, 593–597.
- Landmann, G.; Dumat, W.; Egloff, N.; Gantenbein, A.R.; Matter, S.; Pirotta, R.; Sándor, P.S.; Schleinzer, W.; Seifert, B.; Sprott, H.; et al. Bilateral Sensory Changes and High Burden of Disease in Patients with Chronic Pain and Unilateral Nondermatomal Somatosensory Deficits: A Quantitative Sensory Testing and Clinical Study. Clin. J. Pain 2017, 33, 746–755.
In lines 42-45: “ Furthermore, although QST holds promise for advancing toward mechanism-based pain classification and personalized treatment approaches, its clinical implementation remains limited due to the need for more practical, time-efficient, and standardized protocols”. In general, these methods are created primarily for the diagnosis of damage to various modalities of the somatosensory system and for monitoring. An important feature of these methods is that they can differentiate the degree of damage to thick and thin sensory fibers, whereas EMG studies cannot accurately solve this problem since thin fibers cannot be examined using electric current. These methods are rarely used for differentiation of the type of pain. However, they are widely used, especially in the diagnosis of neuropathic pain syndrome in patients with distal polyneuropathy in the absence of pathology from the peripheral nerves of the lower extremities on EMG.
In lines 46-48 : Among its various modalities, electrical threshold testing (ETT)—which includes the evaluation of sensory, motor, and pain thresholds—has emerged as a promising alternative to traditional mechanical or thermal QST methods”. In my opinion, this idea is incorrect, since electrical threshold determination can help in assessing the sensory threshold of an electrical modality and cannot be a promising alternative to traditional mechanical or thermal QST methods.
In lines 45-48 “A particularly relevant but understudied parameter is the recovery time required between consecutive measurements. This factor is critical because repeated electrical stimulations can transiently alter the excitability of neural pathways through mechanisms such as peripheral fatigue, synaptic modulation, or central sensitization”. Provide specific definitions for the following terms: recovery time, peripheral fatigue, synaptic modulation. However, it is important to note that when performing EMG of peripheral nerves in patients with distal polyneuropathy, repeated subsequent impulses are evoked by lower electrical thresholds compared to the initial values ​​at the beginning of stimulation.
The introduction focused on sensory thresholds and said little about motor thresholds.
In lines 54-58 : “ A particularly relevant but understudied parameter is the recovery time required between consecutive measurements. This factor is critical because repeated electrical stimulations can transiently alter the excitability of neural pathways through mechanisms such as peripheral fatigue, synaptic modulation, or central sensitization”. You referred to the source [8] for this information. This source made its conclusions in the treatment of patients with neuralgia using percutaneous and transcutaneous electrical nerve stimulation. Here we are talking about the therapeutic effect of TENS associated with the release of endogenous endorphins and a decrease in peripheral sensitization. This will not help you confirm your hypothesis. Please look for other sources.
In lines 58-63 : “Without a clear understanding of how quickly thresholds return to baseline after stimulation, the validity of repeated-measure designs—commonly used in both research and clinical contexts—may be compromised. This limitation becomes especially relevant in protocols involving pre-post assessments, intra-session monitoring, or longitudinal studies, where short inter-measurement intervals are often applied under the assumption of full threshold recovery.” This phenomenon has not yet been proven and cannot be used as a reason to set the goal of your research. I have a proposal to include it in the purpose.
The sections "Study Design" and "Participants" are described briefly but quite informatively. However, Please indicate the date of the approval document.
When determining the sensory and motor thresholds it is important to indicate the localization of electrical stimulation. When stimulating above the nerve branches, the threshold will undoubtedly be lower. When stimulating above the active muscle points, the motor threshold in these areas also will be lower.
Determining the motor threshold by visual muscle contraction is subjective. It would be better to use EMG control. Add this to the limitations.
Please add the name of the device used for electrical stimulation and its registration number to the materials and methods.
Please indicate the frequency and duration of electrical stimulation. What were the time intervals between the two measurements?
In lines 123-124: “At each time point, two measurements were recorded to evaluate intra-timepoint repeatability.” This phrase is not clear to me. Can you rephrase it in a different way with more explanations? How were the two measurements recorded simultaneously?. It would be better to provide an illustration that clearly describes the course of your experiment.
What units of measurement were used in the tables 2, 3, 4 and diagram 1? Specify the amplitude in mA
Figure 1 is unclear. Diagrams a, b and c show the differences between the two measures of sensory, motor and pain thresholds. However, in each diagram the sensory threshold has different values! Add axis title to diagrams.
In lines 330-339: Finally, the study by Xia et al. (2020), which assessed EST using circular pin electrodes, offers additional insights into inter-session reliability for cutaneous small fiber assessment [14]. Their results indicated moderate reliability (ICC = 0.595; CV ≈ 25%), and importantly, they noted physiological changes in superficial blood flow and skin temperature during the process. These observations point to autonomic changes that could affect short-term stability—a factor addressed directly in the current design by evaluating how quickly thresholds return to stable values after stimulation. Moreover, while Xia et al. concentrated on longitudinal consistency over days, these findings provide operational recovery windows that could improve intra-session protocol design and reduce measurement artifacts due to carryover effects. The authors conducted a CPT measurement was repeated on two separate days with at least one-week interval and it was not similar to your study. Please explain!
In the discussion, comparisons of the work of other authors require more clarification and should be specifically related to the results obtained.
In the limitations, the authors noted that electrical stimulation was performed with electrical impulses with a frequency of 100 Hz, 100 μs. However, according to the TENS classification, this frequency is classified as high-frequency impulses. Correct in materials and methods “low frequency” to “high frequency”.
Conclusions should be shorter and more specific and there is no need to explain the results obtained. Leave that for the discussion section.
In general, I had difficulty understanding the purpose of the authors' work. The topic is very interesting and can be useful for specialists in this field. However, the authors, both in the materials and methods, and in the results, were unable to clearly and intelligibly explain the course of the work and demonstrate the results obtained to the authors.
In addition, it is doubtful that the threshold is registered after 15 seconds with a single stimulation lasting 0 seconds. After 30 seconds - with a double stimulation lasting 1-15 seconds, after 45 seconds - with a triple stimulation lasting 0-15-30 seconds. That is, their conditions are not the same. In addition, it is doubtful that the threshold registration through the 15-second registration occurs after a single stimulation at 0 seconds. At 30 seconds - with a double stimulation at 0 and 15 seconds, At 45 seconds - after a triple stimulation 0-15-30 seconds. etc. That is, at each time point these conditions are not the same.
Author Response
Response to Reviewer 3 Comments
|
||
1. Summary |
|
|
Thank you very much for taking the time to review this manuscript. Please find the detailed responses below and the corresponding revisions/corrections highlighted/in track changes in the re-submitted files.
|
||
2. Questions for General Evaluation |
Reviewer’s Evaluation |
Response and Revisions |
Does the introduction provide sufficient background and include all relevant references? |
Must be improved |
Thank you for your comment. We have followed your suggestions and revised the Introduction to improve the background context and include additional relevant references, ensuring a clearer rationale for the study. |
Are all the cited references relevant to the research? |
Must be improved |
We have revised cited references to ensure its relevancy to the study. |
Is the research design appropriate? |
Must be improved |
Thank you for your comment. We have improved this section to better justify the research design and ensure it is clearly aligned with the study’s objectives. |
Are the methods adequately described? |
Must be improved |
Based on your suggestions and those of the other reviewers, we have revised and clarified the Methods section. We hope the current version provides a clearer and more comprehensive description of the procedures used. |
Are the results clearly presented? |
Must be improved |
We will present the results in a clearer and more structured manner to enhance their clarity and facilitate understanding. |
Are the conclusions supported by the results? |
Must be improved |
Thank you for your observation. Following your suggestions and those of the other reviewers, we have revised the Conclusions section to ensure it is more clearly supported by the results presented. We hope the current version better reflects the scope and findings of the study. |
3. Point-by-point response to Comments and Suggestions for Authors |
||
Comment: The title of manuscripts must be without abbreviations. It is important to indicate in the title that the study was conducted on healthy population. |
||
Response: Thank you for pointing this out. Following the suggestion, we have revised the title to remove abbreviations and to explicitly indicate that the study was conducted in a healthy population. The new title is: "Recovery Time of Electrical Sensory, Motor, and Pain Thresholds: A Pilot Study towards Standardization of Quantitative Sensory Testing in Healthy Population." We hope this modification makes the title clearer and more appropriate. |
||
Comment: In lines 39-40: “However, despite its broad application, QST presents certain limitations related to reliability, standardization, and test duration” I do not understand why there are such limitations. Conducting complex testing of tactile, temperature, pain and vibration sensitivity is at least mandatory when examining patients with distal polyneuropathy. Since each test is responsible for assessing each modality separately. Response: We thank the reviewer for this thoughtful comment and we agree with the importance of QST. In order to avoid misunderstanding, we have restructured the sentence to emphasize the potential of electrical thresholds testing (ETT) as a method within the framework of QST, rather than focusing on its limitations. We hope that this revised wording better reflects the strengths of QST and clarifies the positioning of ETT in this context. |
||
|
||
Comment: In lines 40-42-“Thermal and mechanical tests are prone to contextual influences, participant variability, and sensitization effects, which may compromise result consistency”. Indeed, a number of authors believe that patients with severe pain develop hyposensitivity of the somatosensory system to nonnoxious mechanical stimuli due to central sensitization. This functional hypoesthesia is reversible and regresses after the pain subsides. This phenomenon is called non-dermatomal somatosensory deficit. Restate this information more precisely, with explanations. I have added additional sources that may be useful for your task. This is a very important section, and it makes sense to focus on it.
Response: We thank the reviewer for this valuable observation and we fully agree with the comment. In the revised manuscript, we have reformulated this part of the Introduction to ensure that it is more precise and aligned with the literature. Our intention with the original sentence was to refer to authors who have highlighted certain limitations of thermal and mechanical tests, such as the longer time required for pain pressure threshold (PPT) measurements, their tendency to yield more censored values compared with current perception threshold, and their sensitivity to external factors including the sedative effects of opioids. We have now restructured the paragraph to clarify these points and provide a more accurate explanation in accordance with the bibliography.
Comment: In lines 42-45: “ Furthermore, although QST holds promise for advancing toward mechanism-based pain classification and personalized treatment approaches, its clinical implementation remains limited due to the need for more practical, time-efficient, and standardized protocols”. In general, these methods are created primarily for the diagnosis of damage to various modalities of the somatosensory system and for monitoring. An important feature of these methods is that they can differentiate the degree of damage to thick and thin sensory fibers, whereas EMG studies cannot accurately solve this problem since thin fibers cannot be examined using electric current. These methods are rarely used for differentiation of the type of pain. However, they are widely used, especially in the diagnosis of neuropathic pain syndrome in patients with distal polyneuropathy in the absence of pathology from the peripheral nerves of the lower extremities on EMG. Response: We understand the concern and agree that the original sentence could lead to confusion. For this reason, we have removed it from the Introduction. Our intention is not to diminish the value of traditional QST or EMG, but rather to emphasize the potential of electrical threshold testing (ETT) as a complementary method. ETT is not an EMG study; instead, it relies on cutaneous electrical stimulation to differentiate sensory, motor, and pain-related mechanisms. With this study, our goal is to establish methodological foundations for ETT as an additional tool that may offer certain advantages, as described in the manuscript.
Comment: Among its various modalities, electrical threshold testing (ETT)—which includes the evaluation of sensory, motor, and pain thresholds—has emerged as a promising alternative to traditional mechanical or thermal QST methods”. In my opinion, this idea is incorrect, since electrical threshold determination can help in assessing the sensory threshold of an electrical modality and cannot be a promising alternative to traditional mechanical or thermal QST methods. Response: We thank the reviewer for this important clarification. We agree with the observation and acknowledge that our initial wording was misleading. In the revised manuscript, we have modified the Introduction to avoid presenting ETT as a replacement for traditional mechanical or thermal QST methods. Instead, we now emphasize that ETT should be considered as a complementary approach that may provide additional information. We hope this adjustment resolves the concern and reflects the more accurate positioning of ETT.
Comment: In lines 45-48 “A particularly relevant but understudied parameter is the recovery time required between consecutive measurements. This factor is critical because repeated electrical stimulations can transiently alter the excitability of neural pathways through mechanisms such as peripheral fatigue, synaptic modulation, or central sensitization”. Provide specific definitions for the following terms: recovery time, peripheral fatigue, synaptic modulation. However, it is important to note that when performing EMG of peripheral nerves in patients with distal polyneuropathy, repeated subsequent impulses are evoked by lower electrical thresholds compared to the initial values ​​at the beginning of stimulation. Response: We thank the reviewer for this insightful comment. In the revised manuscript, we have added specific information to clarify these points. Recovery time is now defined as the interval required between consecutive measurements for the thresholds to return to equivalent values. This is explained because various mechanisms (peripheral fatigue, synaptic modulation, or central sensitization) can transiently increase electrical thresholds. For example, following a high-intensity electrical stimulation on the skin (at a painful level), the electrical sensory threshold becomes temporarily elevated, meaning that a higher current intensity is needed for the participant to perceive the stimulus. We believe that these clarifications improve the precision and understanding of the methodology and its physiological basis.
Comment: The introduction focused on sensory thresholds and said little about motor thresholds. Response: We agree with the comment and have revised the Introduction to incorporate information about motor thresholds, ensuring a more balanced presentation of both sensory and motor aspects of ETT.
Comment: In lines 54-58 : “ A particularly relevant but understudied parameter is the recovery time required between consecutive measurements. This factor is critical because repeated electrical stimulations can transiently alter the excitability of neural pathways through mechanisms such as peripheral fatigue, synaptic modulation, or central sensitization”. You referred to the source [8] for this information. This source made its conclusions in the treatment of patients with neuralgia using percutaneous and transcutaneous electrical nerve stimulation. Here we are talking about the therapeutic effect of TENS associated with the release of endogenous endorphins and a decrease in peripheral sensitization. This will not help you confirm your hypothesis. Please look for other sources. Response: We thank the reviewer for this important observation and understand the confusion that the original wording may have caused. In our study, we used a rectangular symmetrical biphasic current at low frequency (100 Hz), which in clinical practice corresponds to high-frequency or conventional TENS. The immediate and short-term analgesic effect of high-frequency TENS is primarily explained by the gate control mechanism, whereby stimulation of fast-conducting fibers inhibits the transmission of nociceptive input from slower fibers. It is also well established that TENS induces accommodation: after a few seconds of stimulation, the perception of current decreases, leading to higher sensory thresholds. A similar phenomenon occurs at the motor level. In addition, the process of temporal summation, in which consecutive stimuli amplify the nociceptive response, further highlights the need for caution when performing two threshold measurements in close succession. For these reasons, and in order to standardize the evaluation protocol, our study was designed to determine the appropriate recovery interval between consecutive measurements.
Comment: In lines 58-63 : “Without a clear understanding of how quickly thresholds return to baseline after stimulation, the validity of repeated-measure designs—commonly used in both research and clinical contexts—may be compromised. This limitation becomes especially relevant in protocols involving pre-post assessments, intra-session monitoring, or longitudinal studies, where short inter-measurement intervals are often applied under the assumption of full threshold recovery.” This phenomenon has not yet been proven and cannot be used as a reason to set the goal of your research. I have a proposal to include it in the purpose. Response: We thank the reviewer for this constructive observation. We agree that our original wording was imprecise. We have revised the paragraph to clarify that the aim of our study is not based on an unproven assumption, but rather on the need to contribute data that may help to standardise electrical QST protocols. Unlike traditional QST modalities, electrical QST remains less studied and lacks standardised measurement procedures. We therefore believe that the information derived from our study could be valuable in addressing this methodological gap.
Comment: The sections "Study Design" and "Participants" are described briefly but quite informatively. However, please indicate the date of the approval document. |
||
Response: We sincerely thank the reviewer for this helpful comment. As suggested, we have revised the Study Design section to include the date of the ethics approval document. We hope this addition clarifies the ethical approval details and improves the transparency of the manuscript.
Comment: When determining the sensory and motor thresholds it is important to indicate the localization of electrical stimulation. When stimulating above the nerve branches, the threshold will undoubtedly be lower. When stimulating above the active muscle points, the motor threshold in these areas also will be lower. Response: We fully agree with the reviewer’s observation and acknowledge that the localization of stimulation can influence the thresholds obtained. In the present study, we did not have a method to determine the exact localization of the nerve branches. However, electrode placement was standardized as specified in the manuscript and in line with previous studies. Specifically, as described: “Electrodes were placed on the anterior surface of the forearm, aligned longitudinally along the wrist flexor muscle group. The distal electrode was positioned 4 cm proximal to the wrist joint line, while the proximal electrode was placed 4 cm distal to the elbow crease.” We hope this clarification, together with the standardized approach used, adequately addresses the reviewer’s concern.
Comment: Determining the motor threshold by visual muscle contraction is subjective. It would be better to use EMG control. Add this to the limitations. Response: We sincerely thank the reviewer for this valuable observation. Following the suggestion, we have added this point to the Limitations section of the manuscript. We fully agree that determining the motor threshold by visual muscle contraction may be considered subjective, and using electromyography would provide a more objective measure. However, in this study we tried to optimize somatosensory evaluation through electrical thresholds, which are more accessible than other techniques, such as EMG or ENG, which are already well established and standardized. We hope that the revised manuscript now adequately reflects this limitation.
Comment: Please add the name of the device used for electrical stimulation and its registration number to the materials and methods. Response: We thank the reviewer for this helpful suggestion. In the revised manuscript, the Materials and Methods section now specifies the device used for electrical stimulation, including its model and registration number. We hope this clarification improves the transparency and reproducibility of the study.
Comment: Please indicate the frequency and duration of electrical stimulation. What were the time intervals between the two measurements? Response: We agree with this comment. In the revised manuscript, we have added details regarding the stimulation parameters, including frequency, pulse width, and the rate of current increase. We have also clarified the timing of the measurements. Specifically, three separate tests were conducted, with 10-minute intervals between them to ensure recovery. Within each test, pairs of measurements were performed at different time points: one measurement was followed immediately by another at 0 seconds, then after a 3-minute pause a measurement was followed by another at 15 seconds, and so forth in a progressive manner. We hope this additional information makes the methodology clearer.
Comment: In lines 123-124: “At each time point, two measurements were recorded to evaluate intra-timepoint repeatability.” This phrase is not clear to me. Can you rephrase it in a different way with more explanations? How were the two measurements recorded simultaneously?. It would be better to provide an illustration that clearly describes the course of your experiment. Response: We thank the reviewer for this valuable feedback. We agree that the original wording could be confusing. In the revised manuscript, we have rephrased the paragraph to clarify the procedure followed at each time point. Additionally, we have included an illustration (Figure 1) that depicts the experimental course in detail. We hope that this clarification and the new figure make the methodology easier to follow.
Comment: What units of measurement were used in the tables 2, 3, 4 and diagram 1? Specify the amplitude in mA. Response: We sincerely thank the reviewer for pointing out this omission. In the revised manuscript, we have specified the units of measurement in all relevant tables and figures, as well as in the main text.
Comment: Figure 1 is unclear. Diagrams a, b and c show the differences between the two measures of sensory, motor and pain thresholds. However, in each diagram the sensory threshold has different values! Add axis title to diagrams. Response: We understand the concern and have revised Figure 2 and its titles. Figure 2a illustrates the results of the sensory threshold test (STT, with EST as the only outcome), Figure 1b the motor threshold test (MTT, with EST and EMT as outcomes), and Figure 1c the pain threshold test (PTT, including EST, EMT, and EPT). The differences observed between the paired measurements (A–B) reflect the recovery dynamics of the nerve fibers. EST should be different in each diagram as they are different tests performed. In MTT, when the second measurement is performed immediately after the first (0 s), the fibers have not fully recovered, resulting in a higher threshold in the second measurement, as a stronger current is required to depolarize a recently stimulated fiber. In the PTT, variations in the EST also follow this principle, with a trend toward lower differences as the interval between stimuli increases. These findings are explained in more detail in the Results section. We hope that the clarifications provided make the results easier to follow.
Comment: In lines 330-339: Finally, the study by Xia et al. (2020), which assessed EST using circular pin electrodes, offers additional insights into inter-session reliability for cutaneous small fiber assessment [14]. Their results indicated moderate reliability (ICC = 0.595; CV ≈ 25%), and importantly, they noted physiological changes in superficial blood flow and skin temperature during the process. These observations point to autonomic changes that could affect short-term stability—a factor addressed directly in the current design by evaluating how quickly thresholds return to stable values after stimulation. Moreover, while Xia et al. concentrated on longitudinal consistency over days, these findings provide operational recovery windows that could improve intra-session protocol design and reduce measurement artifacts due to carryover effects. The authors conducted a CPT measurement was repeated on two separate days with at least one-week interval and it was not similar to your study. Please explain! Response: We thank the reviewer for this observation and fully agree with the comment. Indeed, our study differs from Xia et al. (2020) in its methodological focus. To our knowledge, no previous studies have specifically examined immediate recovery times between consecutive electrical threshold measurements. The closest available evidence relates to test–retest reliability across different time frames. For this reason, we cited several studies—including Xia et al.—that explored inter- and intra-day reliability of electrical thresholds. While our design is also based on repeated measurements, it specifically addresses much shorter recovery intervals (up to two minutes between stimulations). We are aware that the two approaches are not directly comparable. Therefore, we have revised the manuscript to clarify that these references are not intended as direct comparisons, but rather as examples of related study designs that have contributed to the broader process of ETT standardization.
Comment: In the discussion, comparisons of the work of other authors require more clarification and should be specifically related to the results obtained. Response: We thank the reviewer for this helpful comment. We have revised the Discussion to improve clarity and coherence. As noted, direct comparisons with our findings are not possible because, to our knowledge, no previous studies have applied a comparable methodology to assess immediate recovery times of electrical thresholds. Nonetheless, we have cited studies that provide relevant support in two ways: (i) articles that describe neurophysiological mechanisms potentially underlying our observations, and (ii) studies demonstrating the reliability and consistency of electrical thresholds under different conditions. We hope that these modifications make the discussion clearer and more strongly contextualise our results within the existing literature.
Comment: In the limitations, the authors noted that electrical stimulation was performed with electrical impulses with a frequency of 100 Hz, 100 μs. However, according to the TENS classification, this frequency is classified as high-frequency impulses. Correct in materials and methods “low frequency” to “high frequency”. Response: We sincerely thank the reviewer for this thoughtful observation. We acknowledge that within the specific TENS classification, a frequency of 100 Hz is commonly referred to as “high-frequency TENS.” At the same time, according to the broader classification of electrotherapy currents (as described, for example, in Electrotherapy in Physiotherapy, Editorial Panamericana), currents between 0 and 1000 Hz are generally categorized as “low-frequency currents,” which includes all types of TENS. To avoid any confusion between these two perspectives, we have decided not to use the term “TENS” in the manuscript. Instead, we now refer to the stimulation as “low-frequency symmetrical biphasic current,” which we believe provides a clearer and more precise description.
Comment: Conclusions should be shorter and more specific and there is no need to explain the results obtained. Leave that for the discussion section. Response: We thank the reviewer for this suggestion. In the revised manuscript, we have shortened the Conclusions section, ensuring that it is more specific and focused. The explanation of the results has been moved to the Discussion section, as recommended. We hope that this adjustment improves the clarity and structure of the manuscript.
Comment: In general, I had difficulty understanding the purpose of the authors' work. The topic is very interesting and can be useful for specialists in this field. However, the authors, both in the materials and methods, and in the results, were unable to clearly and intelligibly explain the course of the work and demonstrate the results obtained to the authors. Response: We thank the reviewer for this important observation, and we fully understand the concern. It was not our intention to make the aim of the study unclear, and we regret that it may have been misunderstood. The main objective of our work was to investigate the time required for tissue recovery before performing a new somatosensory evaluation, ensuring that the results are not affected by stimulus summation. We have revised the manuscript to clarify the description of both the study design and the results. In addition, we have included Figure 1, which graphically illustrates the methodology followed and the number of measurements performed. We hope that this revision helps to better align the study objectives with the methodology and makes the course of the work easier to understand.
Comment: In addition, it is doubtful that the threshold is registered after 15 seconds with a single stimulation lasting 0 seconds. After 30 seconds - with a double stimulation lasting 1-15 seconds, after 45 seconds - with a triple stimulation lasting 0-15-30 seconds. That is, their conditions are not the same. In addition, it is doubtful that the threshold registration through the 15-second registration occurs after a single stimulation at 0 seconds. At 30 seconds - with a double stimulation at 0 and 15 seconds, At 45 seconds - after a triple stimulation 0-15-30 seconds. etc. That is, at each time point these conditions are not the same. Response: We thank the reviewer for this thoughtful comment and understand that our original description of the methodology may have caused confusion. We would like to clarify that the thresholds were always measured in pairs, not as accumulations of consecutive stimulations. Specifically, two immediate measurements (with 0 s between them) were first performed. After a 3-minute rest, another pair of measurements was taken with a 15-second interval between them. Following another 3-minute rest, two further measurements were taken with a 30-second interval. After a final 3-minute rest, a last pair of measurements was recorded with a 1-minute interval. This sequence was applied in both the sensory threshold test (Test 1) and the motor threshold test (Test 2). For the pain threshold test (Test 3), two additional pairs were included at 90 and 120 seconds. Regarding the stimulation time, we have clarified in Methodssection that the intensity of the current was increased with a rate of 1 mA per second, until the corresponding thresholds were reached. We hope that this clarification resolves the misunderstanding and makes the experimental procedure easier to follow.
|
||
|
||
4. Response to Comments on the Quality of English Language |
||
The English is fine and does not require any improvement. |
||
|
||
5. Additional clarifications |
||
No additional clarifications were required. |
Round 2
Reviewer 3 Report
Comments and Suggestions for Authors
First of all, thank you very much for the opportunity to review this manuscript again.
I believe this research is very important and has high scientific significance. I truly enjoyed reading this article and the author's responses to my comments. The authors have made a tremendous effort to improve the manuscript.
The changes to the title now better align with the content.
Additional fragments and a revised introduction have improved this section.
The added information in the materials and methods has added more clarity to the work carried out.
Changes in the study results increased the reliability of the results obtained.
The change in discussion with explanations makes it easier for the reader to understand the essence of the work and the clarity and reliability of the results obtained.
The discussion in the revised version is more specific and more relevant to the results obtained and the stated purpose.
In general, I understand that this is a pilot study and this manuscript is only a step toward subsequent work. This happens when a topic hasn't been sufficiently explored and the authors are trying to advance an idea or hypothesis that could become a cornerstone of future research in this area. The authors answered all my questions and explained their positions logically, scientifically and convincingly.